# Changes of Volatile Flavor Compounds in Large Yellow Croaker (*Larimichthys crocea*) during Storage, as Evaluated by Headspace Gas Chromatography–Ion Mobility Spectrometry and Principal Component Analysis

**DOI:** 10.3390/foods10122917

**Published:** 2021-11-25

**Authors:** Tengfei Zhao, Soottawat Benjakul, Chiara Sanmartin, Xiaoguo Ying, Lukai Ma, Gengsheng Xiao, Jin Yu, Guoqin Liu, Shanggui Deng

**Affiliations:** 1Zhejiang Provincial Key Laboratory of Health Risk Factors for Seafood, Collaborative Innovation Center of Seafood Deep Processing, College of Food and Pharmacy, Zhejiang Ocean University, Zhoushan 316022, China; Zhaotengfei1996@yeah.net (T.Z.); dengshanggui@163.com (S.D.); 2International Center of Excellence in Seafood Science and Innovation, Faculty of Agro-Industry, Prince of Songkla University, Hat Yai, Songkhla 90110, Thailand; soottawat.b@psu.ac.th; 3Department of Agriculture, Food and Environment (DAFE), Pisa University, Via del Borghetto, 80, 56124 Pisa, Italy; chiara.sanmartin@unipi.it; 4College of Biosystems Engineering and Food Science, Zhejiang University, Hangzhou 310058, China; 5Guangdong Provincial Key Laboratory of Lingnan Specialty Food Science and Technology, College of Light Industry and Food, Zhongkai University of Agriculture and Engineering, Guangzhou 510225, China; Gshxiao@aliyun.com; 6Academy of Contemporary Agricultural Engineering Innovations, Zhongkai University of Agriculture and Engineering, Guangzhou 510225, China; 7Longyou Aquaculture Development Center, Agricultural and Rural Bureau of Longyou County, Quzhou 324000, China; Z18858398133@163.com; 8School of Food Science and Engineering, South China University of Technology, Guangzhou 510640, China; guoqin@scut.edu.cn

**Keywords:** large yellow croaker (*Larimichthys crocea*), electronic nose, gas chromatography–ion mobility spectrometry, peroxidation value, volatile organic compounds

## Abstract

The large yellow croaker is one of the most economically important fish in Zhoushan, Zhejiang Province, and is well known for its high protein and fat contents, fresh and tender meat, and soft taste. However, the mechanisms involved in its flavor changes during storage have yet to be revealed, although lipid oxidation has been considered to be one important process in determining such changes. Thus, to explore the changes in the flavor of large yellow croaker fish meat during different storage periods, the main physical and chemical characteristics of the fish meat, including the acid value, peroxide value, *p*-anisidine value, conjugated diene value, and identities of the various flavor substances, were investigated and analyzed by multivariable methods, including headspace gas chromatography–ion mobility spectrometry (GC-IMS) and principal component analysis (PCA). It was found that after 60 d storage, the types and contents of the aldehyde and ketone aroma components increased significantly, while after 120 d, the contents of ketones (2-butanone), alcohols (1-propanethiol), and aldehydes (*n*-nonanal) decreased significantly. More specifically, aldehyde components dominated over ketones and lipids, while the *n*-nonanal content showed a downward trend during storage, and the 3-methylbutanol (trimer), 3-methylbutanol (dimer, D), 3-pentanone (D), and 3-pentanone (monomer) contents increased, whereas these compounds were identified as the key components affecting the fish meat flavor. Furthermore, after 120 d storage, the number of different flavor components reached its highest value, thereby confirming that the storage time influences the flavor of large yellow croaker fish. In this context, it should be noted that many of these compounds form through the Maillard reaction to accelerate the deterioration of fish meat. It was also found that after storage for 120 d, the physical indices of large yellow croaker meat showed significant changes, and its physicochemical properties varied. These results therefore demonstrate that a combination of GC-IMS and PCA can be used to identify the differences in flavor components present in fish meat during storage. Our study provides useful knowledge for understanding the different flavors associated with fish meat products during and following storage.

## 1. Introduction

The large yellow croaker (*Larimichthys crocea*) is a prized commercial fish in China and is considered one of the traditional “four major marine products.” This fish is distributed over three geographical locations in China, namely in the East China Sea (the northern part of the Yellow Sea including Lu Yanyu, Yushuyang, Lushan fishery, and other waters), the Guangdong group (mainly in the Taiwan Strait), and the western group (the South China Sea between the Qiongzhou Straight and the mouth of the Pearl River). The meat of the large yellow croaker is nutritious and contains a variety of unsaturated fatty acids and amino acids, its texture is delicate and crisp, and its taste is delicious [1,2,3]. As a result, this fish is a popular seafood among consumers.

In terms of the analyses of flavor compounds in such foods, current analytical techniques, such as gas chromatography (GC) and GC-mass spectrometry (MS), require solid-phase microextraction (SPME) and/or post-treatments [4,5]. By contrast, headspace-gas chromatography–ion mobility spectrometry (HS-GC-IMS) requires no pretreatment, as the solid, liquid, or headspace gas samples are injected directly [6,7]. This method also displays a particularly high sensitivity (at the ppb level) [8,9] and fast detection [10]. Therefore, GC-IMS is promising for identifying product varieties, carrying out quality control, monitoring product freshness, and estimating product shelf lives [11,12,13]. Previously, HS-GC-IMS and HS-SPME-GC-MS have been used to reveal the fingerprints and changes in the aroma of soybeans during fermentation, wherein 115 volatile organic compounds were successfully identified. In addition, through the fingerprint analysis of olive oil by GC-IMS, 39 volatile organic compounds were found to change significantly during fermentation, thereby confirming that IMS can identify different food flavor components [14]. It has also been suggested that GC-IMS could be employed to examine the changes in lipid flavor compounds during the processing of meat, and it was found that the volatile organic compounds (alcohols, aldehydes, ketones, heterocyclic compounds, aromatic hydrocarbons, and esters) present in different meat products can be identified through fingerprinting by such techniques [15,16,17]. However, few reports have been published on the organic volatile fingerprints of large yellow croaker meat after different storage periods [18].

Thus, we herein report the use of GC-IMS to identify changes in the volatile flavor substances of large yellow croaker meat during its oxidation under storage. By measuring the acid value, the peroxide value, the *p*-anisidine value, and the conjugated diene value of the fish meat, its oxidative deterioration is evaluated, and the fingerprint of volatile flavor compounds is established. Ultimately, we aim to provide information to permit exploration of the changes in volatile flavor substances in large yellow croaker meat during storage for different periods of time and to provide theoretical support for its quality control during storage.

## 2. Materials and Methods

### 2.1. Fish Samples

The large yellow croaker samples used in this experiment were provided by the Ocean Family Fishery (Zhoushan Xincheng, Zhejiang, China) and were transported to the Seafood Health Risk Factors Laboratory at Zhejiang Ocean University later. The fish samples had a body length of 19.40–24.00 cm and a weight range of 270.50–320.70 g. All samples were stored at −18 °C for 2 d after capture (the minimum time required to reach land). The yellow croaker fish selected for the purpose of this study originated from Zhoushan (Zhejiang), in the Zhoushan Islands. The dorsal and upper sides of the fish are yellowish brown, the lower and ventral sides are golden yellow, and the head is large. This fish is rich in proteins, trace elements, and vitamins, which are important in the context of human nutrition. In addition, the salt-soluble protein content can reach 11.90 g/100 g, the water-soluble protein content can reach 14.27 g/100 g, the water content can reach 61.85 g/100 g, and the ash content can reach 0.80 g/100 g.

### 2.2. Reagents and Chemicals

The following chemicals were purchased from Sinopharm Chemical Reagent Co., Ltd. (Shanghai, China): ethanol, petroleum ether, sodium thiosulfate, anhydrous sodium carbonate, acetic acid, isooctane, *n*-hexane, trichloromethane, sodium sulfate, *n*-heptane, potassium iodide, phenol, phenolphthalein, potassium hydroxide, tetrahydrofuran, and alumina. All other reagents were of analytical grade and were commercially available.

### 2.3. Sample Processing

The fish samples were stored in a freezer at −18 °C (MDF-U53V, Sanyo, Japan) and at a relative humidity of 60–75% for 0, 5, 10, 20, 30, 40, 60, 90, and 120 d. At the desired time, the fish meat was cut and homogenized at 10,000 rpm for 20 s (FJ200-S, Hunan Li Chen Instrument Technology Company, Changsha, China) to produce a slurry, as outlined in Figure 1. To identify the volatile components present in the large yellow croaker fish samples after different storage periods, GC-IMS (see Section 2.6.2) was used to analyze the samples after 0, 60, 90, and 120 d.

### 2.4. Determination of the Lipid Profiles

The official methods of the American Oil Chemists Society (AOCS) were used to determine the acid values (AV) (Ca 5a-40), the peroxide values (POV) (Cd 8b-90), and the *p*-anisidine values (*p*-AV) (Cd 18–90) of the various samples. Determination of the conjugated diene value (CDV) was carried out using a modification of the technique, as described below [19].

#### 2.4.1. Acid Values

The AV can be used as an indicator of the degree of oil deterioration. To prepare the phenolphthalein indicator required for this measurement, phenolphthalein (1.00 g) was accurately weighed (AR224-CN, Electronic Balance, Orhaus Instruments (Shanghai) Co., Ltd., Shanghai, China) and completely dissolved in 95% ethanol with ultrasonication. After transferring to a 100 mL volumetric flask, the volume was made up to capacity with 95% ethanol. A standard solution of 0.01 mol/L potassium hydroxide was prepared by dissolving potassium hydroxide (0.56 g) in deionized water (1000 mL). Fish meat samples (5.00 g) subjected to different storage periods were homogenized (5000 rpm) and placed in a 100 mL conical flask. The hydrolysate was prepared by adding a mixture of petroleum ether and ethanol (2:1 *v/v*, 50 mL) and 2–3 drops of the phenolphthalein indicator. Subsequently, the mixture was titrated with a 0.01 mol/L potassium hydroxide standard solution. The AV (mg/g) was calculated according to Equation (1):(1)AV=(v × c × 56.1)/m
where v is the volume of potassium hydroxide solution (mL), c is the concentration of potassium hydroxide (mol/L), m is the quality of the meat sample (g), and 56.1 is the molar mass (g/mol) of potassium hydroxide.

#### 2.4.2. Peroxide Values

The fish meat sample was wiped with absorbent paper to remove water and water-soluble impurities. The fish meat sample (5.00 g) was then added to a 250 mL Erlenmeyer flask and a 3:2 (*v*/*v*) acetic acid–isooctane solution (5 mL) was added and swirled to mix well. Subsequently, a saturated potassium iodide (KI) solution (0.5 mL) was added to the sample and allowed to stand for exactly 1 min. After this time, distilled water (30 mL) was added, and the mixture was swirled to mix well. The sample was then titrated with a 0.005 N sodium thiosulfate solution until light yellow color appeared. Subsequently, the starch indicator solution (0.50 mL, >95% purity) was titrated under agitation until the solution turned colorless. A blank sample was also prepared without the addition of the fish meat sample. The POV (meq/kg) was calculated according to Equation (2):(2)POV=(1000 × v × c)/m
where v is the volume of sodium thiosulfate solution (mL), c is the concentration of the sodium thiosulfate standard solution, and m is the mass of fish meat (g).

#### 2.4.3. *p*-Anisidine Values

The fish meat sample was wiped with absorbent paper, and the *p*-anisidine solution was prepared by adding *p*-anisidine (0.125 g) to glacial acetic acid (50 mL). A sample (2.00 g) of the fish meat was then added to a 25 mL volumetric flask, and isooctane (25 mL) was added prior to swirling to mix well. The absorbance (A_0_) of the resulting solution was measured at 350 nm using a spectrophotometer (UV-2600, Tianjin, China). An aliquot (5 mL) of the solution sample was then pipetted into a test tube, and the p-anisidine reagent (1 mL) was added. After 10 min, the sample absorbance (A1) was measured at 350 nm. For the blank sample, isooctane (5 mL) and the *p*-anisidine reagent (1 mL) were added to a separate test tube. The dimensionless *p*-AV was calculated according to Equation (3):(3)$$p-\mathrm{AV}=25\left(1.2A_{1}-A_{0}\right)/W$$
where W is the weight of the fish meat (g), A_1_ is the absorbance of the fat solution after reaction with the *p*-anisidine reagent, A_0_ is the absorbance of the fat solution, 25 is the sample volume (25 mL), and 1.2 is the correction factor.

#### 2.4.4. Conjugated Diene Values

The fish meat sample was weighed and placed in a small beaker. After the addition of isooctane (5 mL) to dissolve the sample, the volume was adjusted to 50 mL using an additional volume of isooctane. The optical absorption of the obtained solution was then measured at 232 nm (UV-2600, Yerco Instrument Co., Ltd., Shenzhen, China) using isooctane as the zero reference. The CDV was calculated according to Equation (4):(4)CDV=AqC × 1
where Aq is the absorbance of the sample at 232 nm, C is the concentration of the sample (g/100 mL) (the solute is the fish meat and the solvent is isooctane), and 1 (cm) is the length of the quartz cell.

### 2.5. Aroma Detection at Different Points during Storage

For sample pretreatment, the fish meat (10.00 g) was sealed in a capped bottle and allowed to stand for 30 min to allow the volatile compounds to equilibrate in the air. An electronic nose (PEN3, Airsense, Berlin, Germany) was employed under previously reported conditions (*n* = 3) [20]. The detection time was 200 s, the sensor cleaning time was 300–500 s, and the data acquisition time was 199–200 s. The various sensors are described in Appendix A.

### 2.6. Volatile Compound Analysis at Different Points during Storage

#### 2.6.1. Sample Processing

After thawing at 4 °C in a refrigerator, the fish meat sample (5.00 g) was placed in a 20 mL headspace bottle equipped with a magnetic screw seal and incubated 40 °C for 20 min. GC-IMS analysis was then carried out in triplicate for each sample.

#### 2.6.2. Headspace Gas Chromatography–Ion Mobility Spectrometry (GC-IMS)

These analyses were performed on a FlavourSpec^®^ GC-IMS system (G.A.S Company, Berlin, Germany) equipped with a 490 micro gas chromatograph (Agilent, Palo Alto, CA, USA), an autosampler (Solid-Phase Microextraction, 57330-U, Supelco, PA, USA), and a headspace sampling unit (Supelco, PA, USA). The GC was equipped with an FS-SE-54-CB-1 capillary column (15 m × 0.53 mm) (Nicolet, PA, USA). The samples in the headspace vials were incubated at 70 °C for 20 min, and after this time, a sample (500 μL) of the headspace was injected automatically (80 °C, splitless mode) via a heated syringe at 50 °C. The flow of the carrier gas was programmed as follows: 2 mL/min for 0–2 min, 30 mL/min for 1–10 min, 100 mL/min for 10–20 min, and 130 mL/min for 20–45 min. The analytes were eluted and separated at 40 °C, then ionized in the IMS ionization chamber by a 3H ionization source (300 MBq activity) in the positive ion mode. The 9.8 cm drift tube was operated at a constant voltage (5 kV) at 45 °C with a nitrogen flow of 150 mL/min. Each spectrum was reported as an average of 12 scans. The syringe was automatically flushed with a stream of nitrogen for 30 s before each analysis and 5 min after each analysis to avoid cross contamination. The retention index (RI) of each compound was calculated using *n*-ketones C_4_–C_9_ (Sinopharm Chemical Reagent Beijing Co., Ltd., Beijing, China) as external references and the calculations were performed by the automated mass spectral deconvolution and identification system. The identification of volatile compounds was performed by comparing the RIs and drift times, and the content of volatile compounds was quantified based on the HS-GC-IMS peak intensity.

### 2.7. Data Processing

All measurements were carried out three times in parallel, and Origin software (OriginLab Corporation, Northampton, MA, USA) was used for data analysis. The values are reported as “mean ± standard deviation” and were analyzed using SPSS 24.0 software (Chicago, IL, USA). The multi-comparison was obtained at *p* < 0.05. The structures of the volatile compounds were identified based on the built-in IMS database. PCA was carried out using the dynamic PCA plug-in program.

## 3. Results and Analysis

### 3.1. Effect of the Storage Time on Lipid Oxidation in Large Yellow Croaker

Four indicators (i.e., the AV, POV, *p*-AV, and CDV) were used to assess the degree of oxidation in the meat of large yellow croaker during storage. As shown in Figure 2, all four values increased significantly during storage (*p* < 0.05). In the fresh fish (0 d), the AV, POV, *p*-AV, and CDV were 2.02 ± 0.12 mg/g, 4.51 ± 0.51 meq/kg, 0.40 ± 0.04, and 12.43 ± 0.56, respectively, thereby indicating that the fresh fish meat contains a rich variety of lipid products [21]. After 90 d, these values increased to 5.02 ± 0.11 mg/g, 18.55 ± 0.64 meq/kg, 1.47 ± 0.04, and 24.59 ± 0.78, respectively. In particular, the POV increased almost fourfold during this storage period, likely due to hydrogen peroxide being the main product of fish lipid oxidation [22,23]. As the storage period was lengthened, fat oxidation in the fish meat led to a further increase in the POV. More specifically, after 120 d, the AV, POV, p-AV, and CDV were 7.62 ± 0.14 mg/g, 23.67 ± 0.55 meq/kg, 2.33 ± 0.03, and 33.57 ± 0.89, respectively. This observed increase in the AV is also consistent with previous studies [24,25] and is closely related to the free fatty acid content [26,27]. Previously, it was reported that the oxidation of fish meat during storage causes the breakage of ester bonds, which releases a large amount of free fatty acids and increases the AV [28]. In terms of the *p*-AV, this value represents the content of secondary products in the fish meat, including aldehydes, ketone alcohols, and acids [29]. It was found that after 120 d, the *p*-AV approached the POV, thereby confirming an increased content of oxidation products in the fish sample [30]. In this context, it has been reported that *p*-AV increases with time due to the decomposition of secondary oxidation products [31]. As outlined in Table 1, the hexanal (M), hexanal (D), and benzaldehyde contents increased significantly from 453.89, 237.69, and 232.42 mg at day 0 to 765.02, 316.81, and 400.98 mg, respectively, after 120 d. Similarly, the contents of ketones 3-octanone, 2-heptanone (M), and 2-heptanone (D) increased from 197.55, 226, and 33.98, at day 0 to 549.19, 770.82, and 129.45, respectively, after 120 d. These observations suggest that after storage for 120 d, the fish meat had undergone a significant degree of spoiling, which would likely have a detrimental effect on its flavor. It should be noted here that the letters M, D, and T in parentheses after a substance name represent the monomer, dimer, and trimer of the substance, respectively.

### 3.2. Identification of Volatile Compounds in Large Yellow Croaker during Storage

To gain further insight into the volatile compounds present in large yellow croaker after different storage periods, GC-IMS was employed to identify the compounds by their retention times in the GC column and their ion migration times during IMS [32]. The results are displayed in Figure 3, wherein a total of 31 peaks, including 8 ketones, 6 alcohols, 12 aldehydes, 2 esters, 2 alkanes, and 1 amine, can be observed. Some other additional signals were also observed. As can be seen from Figure 3, dimers (D) and trimers (T) can also form from monomeric species (M), such as in the cases of 2-heptanone, 3-hydroxy-2-butanone, 3-pentanone, 3-methylbutanol, hexanal, 2-methylbutanal, methylpropanal, and ethyl acetate, which were observed as both monomers (M) and dimers (D), while 3-methylbutanol was also observed as a trimer (T). The analyzed large yellow croaker meat was, therefore, determined to contain the following flavor components: 8 ketones, namely 3-octanone, 2-heptanone (M), 2-heptanone (D), 3-hydroxy-2-butanone (M), 3-hydroxy-2-butanone (D), 3-pentanone (M), 3-pentanone (D), 2-butanone; 6 alcohols, namely 1-hexanol, 3-methylbutanol (D), 1-penten-3-ol, 1-propanethiol, 3-methylbutanol (T), and ethanol; 14 aldehydes, namely hexanal (M), hexanal (D), benzaldehyde, *n*-nonanal, 3-methylbutanal (M), 3-methylbutanal (D), 2-methylbutanal (M), 2-methylbutanal (D), (*E*)-2-pentenal (M), 2-hexenal (M), methylpropanal (M), methylpropanal (D), heptanal (M), and (*Z*)-4-heptenal; 2 esters, namely ethyl acetate (M) and ethyl acetate (D); and 1 amine, namely trimethylamine. Therefore, when passing through the drift region, multiple signals can be observed for a single compound due to the formation of adducts between the analyzed ions and neutral molecules.

Figure 4 shows a plot of the signal strengths measured by the different sensors of the electronic nose during evaluation of the fish meat samples. With a prolonged storage time, the signals from the W3C, W6S, W2W, W3S, and W1C sensors became stronger, indicating that the large yellow croaker meat accumulated aromatic compounds and hydrogen peroxide. In contrast, the signal related to the organic sulfide compound decreased between 0 and 90 d, and then rebounded again at 120 d. This resembles the observations reported by Meng in terms of the formation of heterocyclic aromatic compounds and the associated changes in flavor [33]. From the data measured by the W2W and W6S sensors, it is apparent that the fish flavor is largely determined by organic sulfides and hydrides, while amines play a smaller role [34].

### 3.3. Topographic Map of the Volatile Components Present in Large Yellow Croaker at Different Storage Times

To comprehensively explore the volatile compounds present in the fish meat at different storage times, a topographic map was obtained for the normalized GC-IMS data (Figure 5), wherein the red vertical line indicates the reaction ion peak (RIP), and each point on the right-hand side of the RIP represents a different volatile compound that is present in the sample. As can be seen from the figure, intense signals were observed between retention times of 100 and 300 s and drift times of 0.7 and 1.5 s. In the normalized two-dimensional plots, red (blue) indicates an increase (decrease) in the volatile compound concentration compared to the reference [35,36]. It has also been suggested that the drift rate is related to the concentration of such compounds in the fish samples [37].

### 3.4. Variations in the Volatile Flavor Compounds Present in Large Yellow Croaker after Different Storage Periods

To further compare the volatile flavor compounds present in the fish samples after different storage periods, all peaks in the 2D GC-IMS map were analyzed to establish a fingerprint map (Figure 6). In this map, each row displays all signal peaks from the same sample, while each column shows the signal peaks for the same volatile compounds measured in triplicate for each storage time. More specifically, each heat map indicates the content of the given volatile compound [38]. Such fingerprints provide a panoramic view of all the volatile compounds present in the samples after different storage periods, and a number of unidentified substances are also displayed (Arabic numerals 1–8) [39,40,41].

Based on the data presented in Figure 4 and Figure 6, it is apparent that the fish meat of large yellow croaker contains different volatile organic compound fingerprints after the various storage times. For example, compared with the samples stored for 60, 90, and 120 d, a greater aldehyde content was present in the initial sample (see Table 1), which is consistent with the findings of Duan [42]. It should be noted here that aldehydes possess a low flavor threshold in addition to a characteristic fat aroma at low concentrations [43]. In the fresh fish meat (i.e., at day 0), the key volatile organic compounds present (area A, Figure 6) were methylpropanal (D), methylpropanal (M), hexanal (D), hexanal (M), ethanol, 2-hexenal (M), and 2-methylbutanal (D). Upon increasing the storage period to 60 d, the flavor attributed to 1-hexanol was reduced and had essentially disappeared after 90 d. This may be because oxidative deterioration of the meat proteins dissipated the 1-hexanol flavor [44]. Furthermore, after storage for 60 d, the contributions from ethyl acetate (D), ethyl acetate (M), 3-hydroxy-2-butanone (D), and 3-hydroxy-2-butanone (M) to the fish flavor were significant, with 3-hydroxy-2-butanone (M) being the key component affecting the fish meat flavor during this period (see Figure 6). This compound is known to mainly originate from the oxidation of palmitic acid, stearic acid, and oleic acid [45]. Subsequently, it was found that beyond 60 d of storage, the main flavor components of the large yellow croaker meat were ethyl acetate (D) and ethyl acetate (M), which have a fat-like flavor. However, the oxidation of such compounds produces a pungent smell, similar to that of rotten eggs [46]. In addition, the contributions by the 3-methylbutyraldehyde (D) and 3-methylbutyraldehyde (M) components were more significant after 90 and 120 d, with the highest content being reached after 120 d. It should be noted here that when humans consume fish meat at this stage of oxidation, they will experience chest tightness, nausea, vomiting, fatigue, and weakness, as previously reported [47]. Although aldehydes are the most abundant compounds in pickled products, if their content is too high, a strong pungent smell is produced along with a greasy and oily/waxy odor, and the consumption of such compounds can be detrimental to human health. Our results, therefore, clearly indicate that after 120 d storage, the spoilage of fish meat takes place, leading to increased ketone and alcohol contents [48]. As described previously, short-chain aldehydes can interact with protein aggregates to produce a fatty flavor [49]. After a period of storage, putrefaction can take place, which can produce a rancid smell due to the production of such volatile aldehydes during the microbial degradation of free cysteine and methionine in the fish muscle [50]. Ketones also play a similar role in the flavor deterioration of large yellow croaker meat due to their increased production by lipid oxidation and microbial degradation [51].

### 3.5. PCA of the Characteristic Flavor Compounds Present in Large Yellow Croaker

Finally, delving deeper into the GC-IMS and GC-MS results, we performed PCA based on the fingerprint map of the volatile organic compounds (Figure 7). This method generates principal components that are linear combinations of the input variables, and it is effective for reducing the number of variables and removing abnormal data. According to the PCA results, the accumulative variance contribution rate of PC-1 (61%) and PC-2 (26%) was 87%, thereby indicating that it was facile to distinguish between the large yellow croaker meat samples from different storage periods, as can be seen in the figure. As outlined in Table 1, some volatile organic compounds (e.g., 3-hydroxy-2-butanone (M), 3-hydroxy-2-butanone (D), 3-pentanone (D), ethanol, and trimethylamine) were more common prior to storage, while others (i.e., 3-octanone, 2-heptanone (M), 3-pentanone (M), and 3-methylbutanol (D)) were present primarily beyond 60 d of storage, and some compounds (i.e., 3-pentanone (D), benzaldehyde, (*Z*)-4-heptenal, and trimethylamine) dominated after 120 d. These results are in agreement with the fingerprint map analysis presented in Figure 6. As shown in the PCA plot of the volatiles identified by GC-MS (Figure 7), PC-1 had a remarkable influence on the large yellow croaker meat samples over the various storage periods, and the day 0 samples differed significantly from those at 60 d (*p* < 0.05). Indeed, this difference increased further when the storage period was prolonged to 90 and 120 d. This gradual change in the flavor components during storage is similar to previously reported findings for salmon fillets [52]. Overall, our results indicate that the flavor characteristics of large yellow croaker meat were successfully established using GC-IMS for the various storage periods. In addition, the clustering of the triplicate data confirms the good reproducibility of this measurement method. Thus, we were able to obtain an improved understanding of lipid oxidation in the meat of large yellow croaker during frozen storage by quantitative analysis of the volatile flavor compounds.

## 4. Conclusions

To explore the changes in flavor of large yellow croaker fish meat during storage, the main physical and chemical characteristics of the fish meat, including the acid value, the peroxide value, the *p*-anisidine value, the conjugated diene value, and the identities of the various flavor substances, were investigated and analyzed by multivariable methods, including headspace gas chromatography–ion mobility spectrometry (GC-IMS) and principal component analysis (PCA). Significant changes were observed in the volatile organic compounds present in the meat samples during storage for periods up to 120 d. A total of 31 volatile compounds were identified by GC-IMS, including aldehydes, ketones, alcohols, esters, and alkanes. Aldehydes accounted for the largest number of these 31 identified species, followed by ketones and alcohols. The GC-IMS data were further used to construct fingerprint maps to highlight the characteristic molecular species present at each stage during storage, and the distinct fingerprint maps were corroborated by results from PCA for the samples at 0, 60, 90, and 120 d. The techniques employed herein could, therefore, be employed to improve quality control and inventory monitoring, as well as for analyzing the flavor components present in other foodstuffs during storage. In order to ensure the smooth progress of this research, our team is still working in relation to this project, I hope you can look forward to our future work report.

## Figures and Tables

**Figure 1 foods-10-02917-f001:**
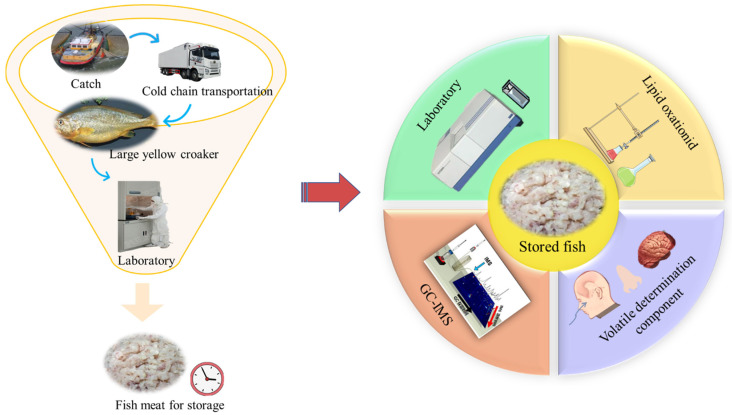
Schematic diagram of the methods employed for analysis of large yellow croaker fish meat.

**Figure 2 foods-10-02917-f002:**
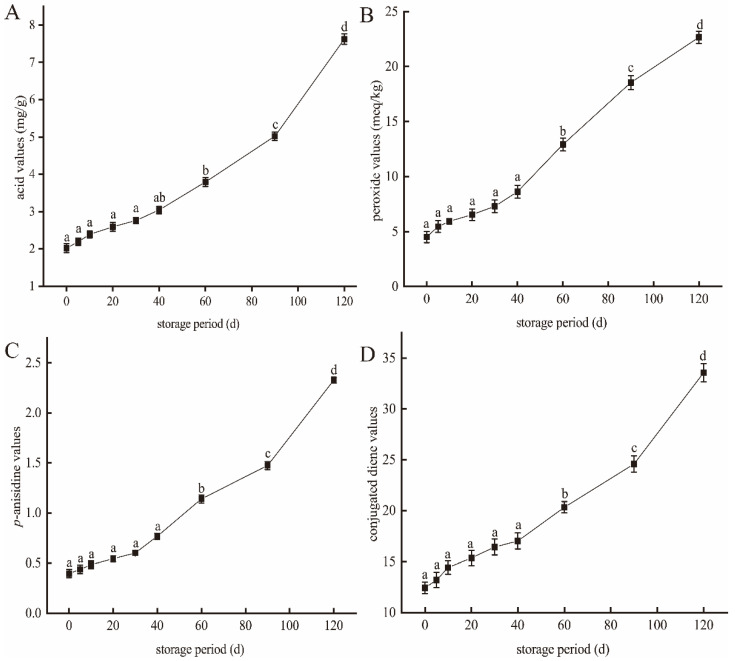
Evaluation of the lipid oxidation taking place in the meat of large yellow croaker fish after different storage periods. Note: (**A**) Acid values, (**B**) peroxide values, (**C**) p-anisidine values, and (**D**) conjugated diene values. The p-anisidine and conjugated diene values are dimensionless. Letters a-d indicate significant differences. The data are presented as means ± SD (*n* = 3).

**Figure 3 foods-10-02917-f003:**
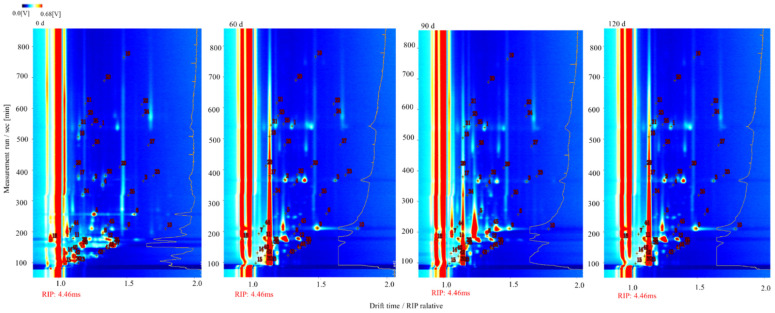
GC-IMS data for the large yellow croaker fish meat after different storage periods (*x*-axis: the ion migration time, *y*-axis: the GC retention time).

**Figure 4 foods-10-02917-f004:**
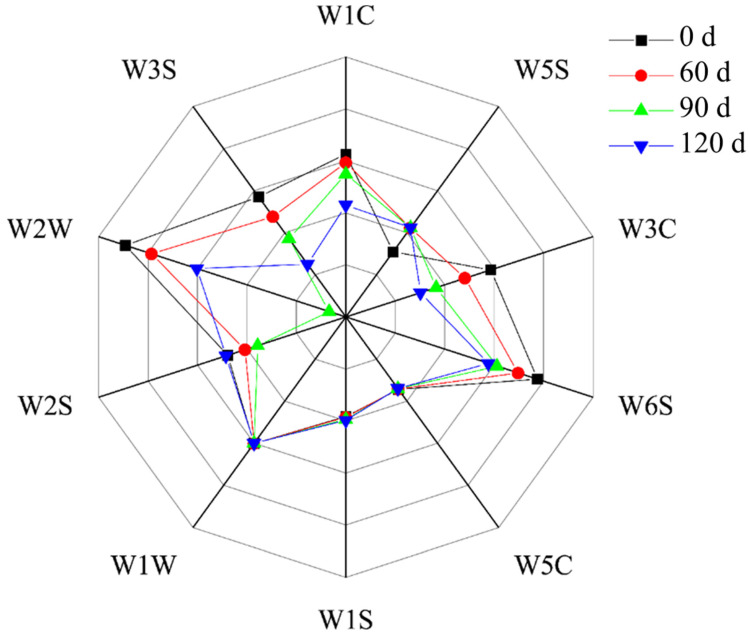
Effect of the storage period on the volatile flavor compounds present in fish meat, as measured using different sensors of an electronic nose. Note: W1C: aromatics; W5S: nitrogen oxides; W3C: ammonia and aromatic components; W6S: hydride; W5C: olefins and aromatic molecules; W1S: methane; W1W: sulfide; W2S: ethanol and some aromatics; W2W: organic sulfides: W3S: alkanes and aliphatics.

**Figure 5 foods-10-02917-f005:**
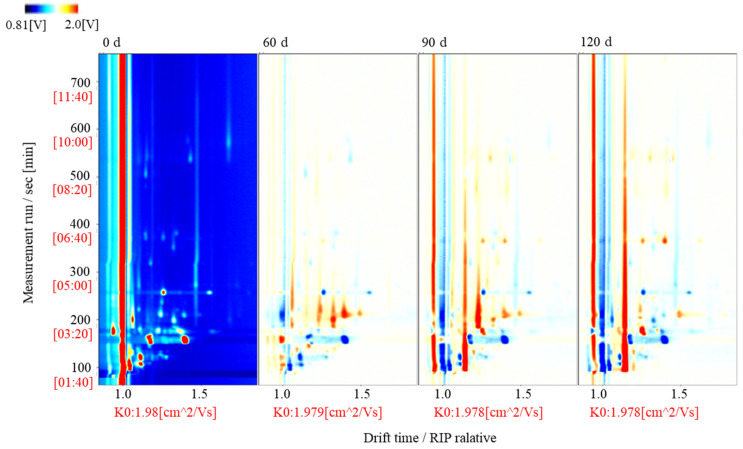
Two-dimensional topographic map of the normalized GC-IMS data for the fish samples subjected to different storage periods.

**Figure 6 foods-10-02917-f006:**
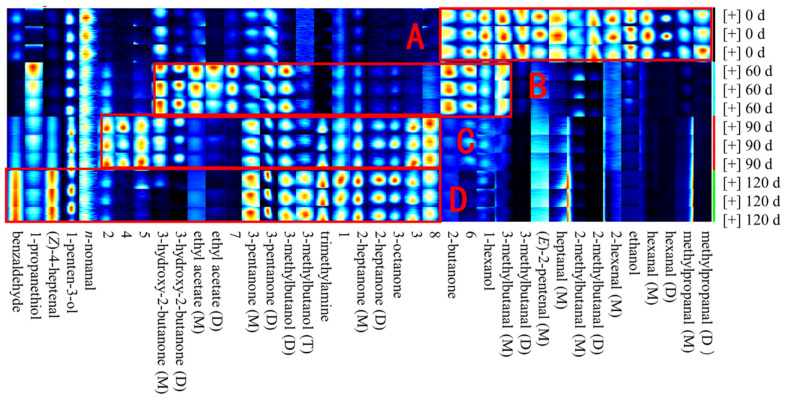
Fingerprint map of the volatile organic compounds present in large yellow croaker after different storage periods. Note: A–D represents that the storage periods of fish meat samples are 0 d, 60 d, 90 d and 120 d, respectively.

**Figure 7 foods-10-02917-f007:**
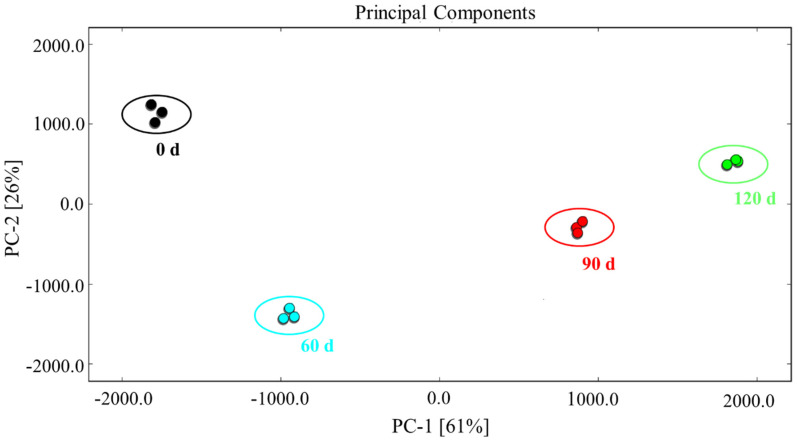
PCA analysis of the characteristic flavor compounds present in large yellow croaker meat after different storage periods.

**Table 1 foods-10-02917-t001:** Qualitative analysis of the flavor compounds present in large yellow croaker meat after different storage periods.

No.	Compound	CAS#	Molecule Formula	MW	RI1	RT2	DT3	Storage Period			
								0 d	60 d	90 d	120 d
1	3-Octanone	C106683	C_8_H_16_O	128.2	992.4	540.9	1.31	197.55	232.44	439.69	549.19
2	2-Heptanone (M)	C110430	C_7_H_14_O	114.2	892	367.6	1.256	226	367.96	572.06	770.82
3	2-Heptanone (D)	C110430	C_7_H_14_O	114.2	889.2	364	1.625	33.98	46.23	99.86	129.45
4	3-Hydroxy-2-butanone (M)	C513860	C_4_H_8_O_2_	88.1	716.1	192.3	1.068	1336.8	1350.1	1362.8	1376.56
5	3-Hydroxy-2-butanone (D)	C513860	C_4_H_8_O_2_	88.1	715.8	192	1.326	884.81	914.77	953.59	987.56
6	3-Pentanone (M)	C96220	C_5_H_10_O	86.1	696.6	178.5	1.108	260.51	339.22	389.51	518.62
7	3-Pentanone (D)	C96220	C_5_H_10_O	86.1	694.1	176.8	1.351	1270.9	1472	1588.4	1631.99
8	2-Butanone	C78933	C_4_H_8_O	72.1	590.9	129.7	1.057	624.73	561.24	483.93	321.34
9	1-Hexanol	C111273	C_6_H_14_O	102.2	879.8	351.9	1.326	225.16	122.06	83.02	94.84
10	3-Methylbutanol (D)	C123513	C_5_H_12_O	88.1	739.9	210.7	1.499	202.46	505.84	820	1088.71
11	1-Penten-3-ol	C616251	C_5_H_10_O	86.1	690.1	174.1	0.944	226.6	248.48	305.92	339.2
12	1-Propanethiol	C107039	C_3_H_8_S	76.2	633.3	147	1.36	109.48	82.32	50.17	47.2
13	3-Methylbutanol (T)	C123513	C_5_H_12_O	88.1	738.8	209.8	1.788	56.03	67.77	97.83	126.9
14	Ethanol	C64175	C_2_H_6_O	46.1	491.5	96.79	1.046	975.52	911.11	711.76	518.53
15	Hexanal (M)	C66251	C_6_H_12_O	100.2	793	257.8	1.264	453.89	534.21	643.29	765.02
16	Hexanal (D)	C66251	C_6_H_12_O	100.2	792.5	257.3	1.56	237.69	256.85	298.56	316.81
17	Benzaldehyde	C100527	C_7_H_6_O	106.1	975.6	507	1.145	232.42	302.87	356.59	400.98
18	*n*-Nonanal	C124196	C_9_H_18_O	142.2	1103	765.3	1.481	147.74	124.97	123.93	110
19	3-Methylbutanal (M)	C590863	C_5_H_10_O	86.1	648.9	153.9	1.169	498.2	419.74	397.01	260.11
20	3-Methylbutanal (D)	C590863	C_5_H_10_O	86.1	646.2	152.7	1.406	157.91	111.17	92.31	56.65
21	2-Methylbutanal (M)	C96173	C_5_H_10_O	86.1	667.7	162.7	1.164	172.2	163.62	154.96	143.92
22	2-Methylbutanal (D)	C96173	C_5_H_10_O	86.1	667.7	162.7	1.4	263.97	231.09	210.64	186.84
23	(*E*)-2-Pentenal (M)	C1576870	C_5_H_8_O	84.1	747.8	217.2	1.104	146.57	132.75	129.54	116.53
24	2-Hexenal (M)	C505577	C_6_H_10_O	98.1	853.6	320.4	1.179	166.87	156.54	146.83	136.59
25	Methylpropanal (M)	C78842	C_4_H_8_O	72.1	568	121.3	1.113	317.71	289.15	254.57	229.72
26	Methylpropanal (D)	C78842	C_4_H_8_O	72.1	570.3	122.1	1.282	214.97	201.43	198.34	187.5
27	Ethyl acetate (M)	C141786	C_4_H_8_O_2_	88.1	609.6	137.1	1.094	184.81	143.91	100.19	58.11
28	Ethyl acetate (D)	C141786	C_4_H_8_O_2_	88.1	611.6	137.9	1.336	120.57	109.78	82.49	53.92
29	Heptanal (M)	C111717	C_7_H_14_O	114.2	902.5	382.7	1.347	55.08	51.23	46.75	39.04
30	(*Z*)-4-Heptenal	C6728310	C_7_H_12_O	112.2	900.6	380.1	1.146	130.31	159.75	189.86	223.49
31	Trimethylamine	C75503	C_3_H_9_N	59.1	496.6	98.26	1.148	1543.4	1665.65	1862.3	1950.94

Note: The retention times and ion migration times are listed together with the compound name, CAS number, molecular formula, molecular weight (MW), reserved index (RI1), retention time (RT2), drift time (DT3), and response peaks after different storage periods.

## Data Availability

The data that support the findings of this study are available from the corresponding author upon reasonable request.

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
