# Peer review of "Changes of Volatile Flavor Compounds in Large Yellow Croaker (Larimichthys crocea) during Storage, as Evaluated by Headspace Gas Chromatography–Ion Mobility Spectrometry and Principal Component Analysis"

_foods, 2021, doi:10.3390/foods10122917_

Round 1

Reviewer 1 Report

Eventhough the manuscript describes a novel hypothesis, the way it is written and data along with the reserch material is presented, is poor. The authors need to seek the help of a native English speaker to proof-read the whole manuscript. I have made an effort to indicate importnat problems within the manuscript. There are more similar issues that must be covered line by line in the manuscript.

COMMENTS

-Abstract

Line 33.''methods''.

Line 37 and eslewhere. Delete''components'', just ''nonanal''.

Line 38. Change ''which'' to ''whereas''.

Lines 40-42. there is no connection in these sentences. Revise.

-Introduction

Lines 66-75. Revise the whole paragraph in a more scientific way.

-Materials and methods

Line 104. There is a problem in this sentence.

-Several parts in the materials and methods must be revised.

-Results and discussion

Line 215. ''during storage for....''.

Line 237-238. Re-organize in a more proper way.

Line 260.''in''.

Line 291.''..of storage...''.

Line 299.''...was lower....''., ''dissapeared...''.

Lines 304-305. Rephrase in proper English.

Line 310. Same comment as above.

Lines 312-328. Revise in a more proper scientific way.

Line 338. This sentence makes no sense. Check it.

Lines 334-358. Revise in proper English.

-Figures

The quality of the GC-IMS figures must be improved.

Author Response

Journal: Foods

Manuscript ID: foods-1453028

Title: Changes of volatile flavor compounds in large yellow croaker (Larimichthys crocea) during storage, as evaluated by headspace gas chromatography-ion mobility spectrometry and principal component analysis

Author(s): Tengfei Zhaoa, Soottawat Benjakulb, Chiara Sanmartinc, Xiaoguo Yinga,d*, Lukai Mae,f*, Gengsheng Xiaoe, Jin Yug, Guoqin Liuh and Shanggui Denga

Dear Prof. Dr. reviewer #1.

Thank you for your waiting. We sincerely apologize. Thanks for the suggestions and repair opportunities given to us by《Foods》journal. Our original intention is to provide food journal with higher quality manuscripts, resulting in the delay of repair manuscripts. We have uploaded all the revised manuscript (foods-1453028). We are willing to cooperate with foods Journal for a long time and look forward to the smooth publication of the manuscript in the journal. All the attachments of the manuscript have been uploaded.

On behalf of my co-authors, we thank you very much for giving us an opportunity to revise our manuscript. We also appreciate you very much for their positive and constructive comments, and suggestions on our manuscript (foods-1453028). We have studied your comments carefully and have made revision in the resubmitted manuscript (marked as red color).

We sincerely looking forward to your reply, thank you.

Yours sincerely

Tengfei Zhao

Corresponding author: Lukai Ma,

Reviewer 2 Report

Revision

Changes of volatile flavor compounds in large yellow croaker (Larimichthys crocea) during storage by headspace gas chromatography-ion mobility spectrometry and principal component analysis

General comment:

The topic of the paper is interesting and the experiment is properly planned. However, there is many doubts  in methodology and interpretation of the results

The detailed comments are listed below:

Abstract

The abstract needs improvement:
1. Line  34: Why do the authors say that they conducted physicochemical studies of fish meat ?

  1. Please delete the sentence: “The results of principal component analysis (PCA) showed that PC-1 was 61% and PC-2 was 26%”. This sentence is not informative for reader.
  2. Line 36: “Aldehydes were more obvious ….” than – please revise the sentence.
  3. Please revise the sentence in lines 40-41
  4. There is a lack a conclusion which derived from the experiment. The last sentence (Line 42-45) is very general. I found such conclusion in many papers. What new is this conclusion bringing ?

Introduction:

Lines 77-79:

Are the Authors sure that they investigated the physical properties in large yellow croaker meat during storage. Please explain which methods were used ?

Lines 80-83:

Please reformulate the sentence is incomprehensible.

Materials and methods:

General comment:

  1. Please explain the source of the analytical methods: acid value, peroxide value, p-Anisidine value, conjugated dienes (also supply the citation in the references). Whether they were ISO, AOCS or other procedures ?
  2. It is unclear whether the analyzes were carried out in meat or in lipids extracted from meat ?
  3. The description of GC_IMS analysis needs supplementation (details about instrument, MS parameters, information about qualitative and quantitative analysis, standards). Please supply.
  4.  There is a mistake, acid value is used for analysis hydrolytic  changes (you can find this information in some references for example  https://doi.org/10.1016/j.lwt.2019.04.067

Results and discussion:

  1. Instead of supplementary material the Figure 1S should be included in the main text manuscript.
  2. Please explain why there were not observed breakdown the PV values and decrease while anisidine value increased and relative content some aldehydes, ketones (measured by GC/MS) which are secondary lipid oxidation products.
  3. Line 195-196. Please revise sentence you are not studied nutritional value of meat you studied the contents of lipid oxidation products in meat (and hydrolytic changes).
  4. The sentence 196-198, please add “respectively” at the end of the sentence.
  5. Line 203: should be “lipid oxidation” (no oil).
  6. Lines 203-204. Why authors write about decomposition of hydroperoxides ?  The not observed such phenomena (I asked about it ) Fig 1 S.
  7. Line 209-210, that is not true  only these carbonyl compounds which react with p-anisidine.
  8. Line 212-213- Please delete redundant sentence.
  9. Lines 214-215: please move this sentence to the Materials nad Methods section.
  10. There is no need to introduce the composition of Table 1  ( CAS number, molecular formula, molecular weight (MW), reserved index 220 (RI1), retention time (RT2), drift time (DT3), and response peaks) , please delete from the text. These information should be placed under the table 1.
  11. There is a lack of discussion in part 3.2. Please supply.
  12. I suggest deleting Figure 4 as it doesn't explain anything.
  13. The authors not explained what the received after PCA analysis of VCs data.

Author Response

Journal: Foods

Manuscript ID: foods-1453028

Title: Changes of volatile flavor compounds in large yellow croaker (Larimichthys crocea) during storage, as evaluated by headspace gas chromatography-ion mobility spectrometry and principal component analysis

Author(s): Tengfei Zhaoa, Soottawat Benjakulb, Chiara Sanmartinc, Xiaoguo Yinga,d*, Lukai Mae,f*, Gengsheng Xiaoe, Jin Yug, Guoqin Liuh and Shanggui Denga

Dear Prof. Dr. reviewer #2.

Thank you for your waiting. We sincerely apologize. Thanks for the suggestions and repair opportunities given to us by《Foods》journal. Our original intention is to provide food journal with higher quality manuscripts, resulting in the delay of repair manuscripts. We have uploaded all the revised manuscript (foods-1453028). We are willing to cooperate with foods Journal for a long time and look forward to the smooth publication of the manuscript in the journal. All the attachments of the manuscript have been uploaded.

On behalf of my co-authors, we thank you very much for giving us an opportunity to revise our manuscript. We also appreciate you very much for their positive and constructive comments, and suggestions on our manuscript (foods-1453028). We have studied your comments carefully and have made revision in the resubmitted manuscript (marked as red color).

We sincerely looking forward to your reply, thank you.

Yours sincerely

Tengfei Zhao

Corresponding author: Lukai Ma,

Round 2

Reviewer 1 Report

In the resubmitted version, there is indeed, a very good article that deserves publication in foods. After my initial rejection (due to numerous problems), now the authors have addressed all my suggestions and therefore I suggest the publication of their article in Foods journal as a fine contribution to the field of this particular research.

Reviewer 2 Report

Dear Authors thank you for diligent work and answers on all comments. I appreciate that. Best regards